# Quantitative Analysis of Melanosis Coli Colonic Mucosa Using Textural Patterns

**Chung-Ming Lo [1,2]**, **Chun-Chang Chen [1]**, **Yu-Hsuan Yeh [1]**, **Chun-Chao Chang [3]** and **Hsing-Jung Yeh [1,3,\*]**

[1] Graduate Institute of Biomedical Informatics, College of Medical Science and Technology, Taipei Medical University, Taipei 110, Taiwan; buddylo@tmu.edu.tw (C.-M.L.); seanchen@tmu.edu.tw (C.-C.C.); a020722115847@gmail.com (Y.-H.Y.)

[2] Graduate Institute of Library, Information and Archival Studies, National Chengchi University, Taipei 116, Taiwan

[3] Division of Gastroenterology and Hepatology, Department of Internal Medicine, Taipei Medical University Hospital, Taipei 106, Taiwan; chunchao@tmu.edu.tw

\* Correspondence: yiew@ms10.hinet.net; Tel.: +886-2-7951-0030

**Abstract:** Melanosis coli (MC) is a disease related to long-term use of anthranoid laxative agents. Patients with clinical constipation or obesity are more likely to use these drugs for long periods. Moreover, patients with MC are more likely to develop polyps, particularly adenomatous polyps. Adenomatous polyps can transform to colorectal cancer. Recognizing multiple polyps from MC is challenging due to their heterogeneity. Therefore, this study proposed a quantitative assessment of MC colonic mucosa with texture patterns. In total, the MC colonoscopy images of 1092 person-times were included in this study. At the beginning, the correlations among carcinoembryonic antigens, polyp texture, and pathology were analyzed. Then, 181 patients with MC were extracted for further analysis while patients having unclear images were excluded. By gray-level co-occurrence matrix, texture patterns in the colorectal images were extracted. Pearson correlation analysis indicated five texture features were significantly correlated with pathological results ($p < 0.001$). This result should be used in the future to design an instant help software to help the physician. The information of colonoscopy and image analystic data can provide clinicians with suggestions for assessing patients with MC.

**Keywords:** gray-level co-occurrence matrix; melanosis coli; colon adenoma

## 1. Introduction

Melanosis coli (MC) is characterized by a brownish-black color change in the colon caused by anthraquinone ($C_{14}H_8O_2$)-containing laxative agents. MC is diagnosed by images from colonoscopy or capsule endoscopy. Patients with MC are likely to experience colon polyps [1] and require close tracking with colonoscopies. The mechanism underlying anthraquinone-containing drug–related MC and colorectal hyperplasia polyps [1,2] involves the conversion of anthraquinone to lipofuscin, which causes mild inflammation in these cells and eventually leading to black or dark brown mucosal appearance on colonoscopy [3].

MC-affected colon mucosal membranes and polyps have special textures, which can be used for imaging analysis. A literature review demonstrated proteomic differences between normal colorectal mucosa and MC with colon cancer [4]. A 2002 study used a computer-assisted diagnosis (CAD) system to assist radiologists in computer tomography colonoscopy to detect colon polyps [5]. Computer-assisted colonoscopy analysis in terms of endoscopy has been published: In 2003 [6], it was

used for colonoscopy video images to label abnormal colorectal mucosa for helping gastroenterologists to diagnose colon polyps and tumors. The paper recorded good results with a sensitivity of 90% and specificity of 97%. The result proves that it is feasible to assist endoscopic physicians with computer-assisted methods.

According to research, CAD analysis of colonoscopy images has been proven to be useful, but no studies have yet used gray-level co-occurrence matrices (GLCMs) or other texture analysis models to analyze images obtained from patients with MC during general endoscopy nor the relationship between mucosal image texture and pathological results in patients with MC. Therefore, this study analyzed the pathological results of polyps of patients with MC through the texture features of general colonoscopy images. The GLCM model can analyze image data suitably [7] and is suitable for MC with special textures. If MC and polyps are detected during colonoscopy, the possible pathological results can be predicted by analyzing images. The patients can be subsequently instructed to stop using anthranoid laxative agents, and a polypectomy is scheduled for poor differentiated polyp which means a better prognosis. [8].

MC is a historical disease [9,10]. Cruveilhier first discovered this phenomenon in 1829, and in 1857, Virchow officially named it MC [11]. Long-term use of anthranoid laxative or of these herbs can cause MC after 3–13 months [12], these drugs include sennoside; Normacol; the Chinese herbal medicine containing polygonum, aloe, senna, and rhubarb; and cascara sagrada [13]. MC can be divided into three phases according to chromaticity and texture [11]: (1) light brown colonic mucosa with no apparent boundaries with normal mucosa, (2) dark brown colonic mucosa with clear linear or noncontinuous boundaries with normal mucosa, and (3) dark black colonic mucosa with linear or spotted boundaries with normal mucosa. MC often disappears after the relevant drug is stopped for six months to one year.

Colon polyps are epidermal protrusions in the lumen of the colon and can be roughly classified into six categories according to the Paris classification: pedunculate, sessile, superficial and elevated, superficial and flat, superficial and depressed, and concave [14]. In general, colorectal polyps can be roughly divided into three categories: (1) hyperplastic polyps, (2) adenoma polyps (adenoma; potentially divided into tubular adenoma, tubulovillous adenoma, and villous adenomas, which is the most malignant), and (3) neoplastic polyps (including carcinoma in situ, carcinoid polyp, and adenocarcinoma). The earlier detection of neoplastic polyps makes the treatment become more effective [8]. Auxiliary methods can help gastroenterologists to gauge which polyps most urgently require treatment. This can solve problems related to malignant adenoma and even colorectal cancer in patients with MC.

In Mainland China, MC is prominent among men aged >60 years and adenomatous polyp hyperplasia is the most common comorbidity [2]. These patients are more prone to proliferative polyps and adenomas [15] and to having higher adenoma detection rates (ADRs) than patients receiving general colonoscopy [1]. MC patient are prone to polyposis; the proportion of patients in the class is approximately 34.7% [1], whereas the corresponding proportion for the normal population is 26.5% [1]. MC prevalence during colonoscopy is 1.78% according to a 2018 Chinese study [2] and approximately 3.13% in a 1993 colonoscopy study [16]. Thus, MC may be related to colorectal cancer. In addition, MC was observed in patients with colorectal cancer [15]. Therefore, MC is caused by the drug causing chronic inflammation and blackening of the colon, and its relationship with carcinoembryonic antigens (CEAs) warrants discussion.

## 2. Materials and Methods

### 2.1. Colonoscopy Database

All collected colonoscopy images were captured with the general colonoscopes produced by Olympus Corporation of Japan (Shinjuku Monolith, 2-3-1 Nishi-Shinjuku, Shinjuku-ku, Tokyo 163-0914, Japan). Colonoscopes models were GF-260 and 290, so we can ensure that the image quality were close.



Endoscopic images were obtained by gastroenterologists from Taipei Medical University Hospital, Taipei, Taiwan.

A small amount of patient data was used for analysis, and decided the next method of analyzing pictures and data for the research. We analyzed data of 298 patients with MC from the database between 1 January 2016 and 17 July 2017; In the initial analysis of the study, the analysis of polyp texture in patients without tumor index such as CEAs or CA 199 was not related to pathological findings, so we included only 124 patients who had CEA data and clear colonoscopy images. We then analyzed these patients' colonoscopy images. The 124 patients' images survey indicated that if analysis is conducted using a large size image or if the reflection of feces are not avoided and the part with white text recorded in the image is not removed, it may be caused by the aforementioned interference. This can lead to unsatisfactory results. Regarding the usefulness of correcting images according to patients' normal skin and luminosity, because the calculation results are based on the gray-level symbiotic matrix analysis method, the final calculated result exerted little effect. The patient's normal skin and brightness were therefore not ultimately required for correction. We also observed that taking a polyp image consisting of $100 \times 100$-pixel squares from the center point provided improved texture features and results.

Next, 51,891 colonoscopy data were acquired from the Division of Gastroenterology and Hepatology, Department of Internal Medicine, Taipei Medical University Hospital. The research encompassed the period from 1 January 2012 to 30 September 2017. The research materials and images were deidentified by the Taipei Medical University Human Research Joint Ethics Committee for research permission and ensured that research materials and images were deidentified. No informed consent is required for this retrospective study. After the adoption, relevant research and statistical work was commenced. Among them, 28,974 colonoscopy data had polyps; thus, the polyp detection rate (PDR) was 55.84%. Because 12,244 times colonoscopy data presented with adenoma or colorectal cancer, adenoma detection rate (ADR) was 23.60%. Finally, 834 patients presented with colorectal cancer, so the detection rate of colorectal cancer was 1.61%.

Next, from these data, the total number of patients with a diagnosis of MC was observed to be 1092, and the MC prevalence rate was 2.1%, similar to the prevalence rate of 1.78% in Mainland China [2]. These patients included 787 women and 305 men, and therefore, the women had an incidence of MC 2.58 times that of men. The age distribution of these patients was as young as 17 and as large as 97. Of these, 96 patients were over 80 years old, and 480 patients were between 60 and 80 years old. Only three patients were younger than 20. The first patient received colonoscopy study on 2 January 2012. The last patient underwent a colonoscopy on 29 September 2017. Among these patients, 658 patients had polyps. 390 patients had colon adenoma detected by colonoscopy. 18 patients' pathology reports had colon cancer.

Based on the data collected from 1092 person-times, the PDR and ADR were 60.26% and 35.71%, respectively—close to the 34.7% noted in other studies [1]. Colorectal cancer detection rate was 1.65%. Compared with data from the previous database, patients with MC had a slightly higher PDR than did patients who had received general colonoscopy (4.42%); moreover, the ADR was much higher (12.11%), and the colorectal cancer detection rate was similar. The result indicating that patients with MC were more prone to adenoma corresponds with results reported previously [1] and indicates that even if patients with MC are more likely to exhibit polyps under colonoscopy, they have a higher adenoma incidence than other people do. MC is associated with adenomas. The study initially used a smaller quantity of data and image analysis to help determine the pattern of screening and analysis for all subsequent data.

We excluded the data of patients without CEAs, poor colon preparation, or with unclear data. Finally, 370 patients were included in the image analysis study. The relationship between image texture GLCMs and CEAs was compared. Among the 370 patients, 181 had polyps and pathological biopsies. The study used images of 181 patients. All pictures collected by one gastroenterologist. Using Image J, a $100 \times 100$-pixel block diagram—taken from outside areas where feces had accumulated or from positions in the centers of polyps—was isolated for analysis. Images were divided into three groups:

1. Cecem, image of the appendix and cecal mucosa (C), consisting of 181 images.
2. Splenic flexure spleen images (S), consisting of 181 images.
3. Polyp (P), if the patient had a polyp or a tumor. If multiple polyps were present, we collected the largest and most prominent textured polyp. The gastroenterologist manually cut the image from central point of the polyp. This also resulted in 181 images. Figure 1a shows one melanosis coli patient's cecal image and Figure 1b shows the same patient's cecal image after stopping anthraquinone containing laxative agents for six months.

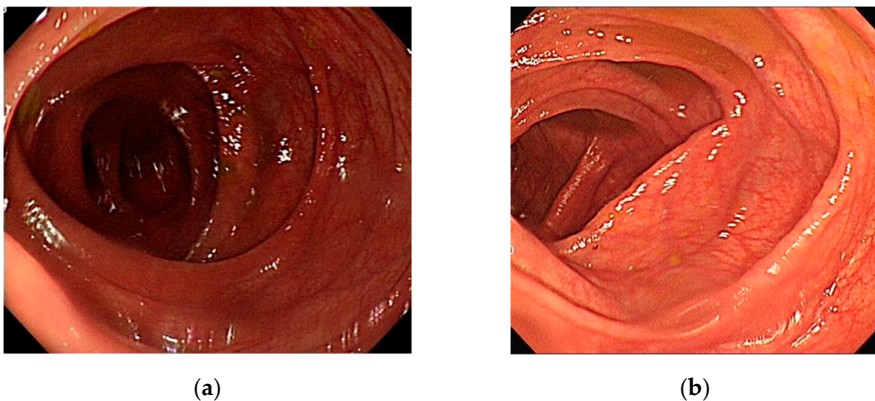

(a)    (b)

**Figure 1.** Melanosis coli patient's cecal images. (**a**) The patient took anthraquinone containing laxative medicine one year and his colonoscopy image revealed blackish mucosal pattern in colon. (**b**) The same patient's cecal image after six months. Blackish mucosal pattern disappeared after stopping anthraquinone containing laxative agent for 6 months.

This section describes the image analysis methods used in this study, like this article [17]. The way we use computer programs to analyze textures has also been widely used in other areas [18]. The images are divided into three groups as previously described. Example images of patients with MC are illustrated in Figure 2a–c and the examples of regions of interest in Figure 2d–f.

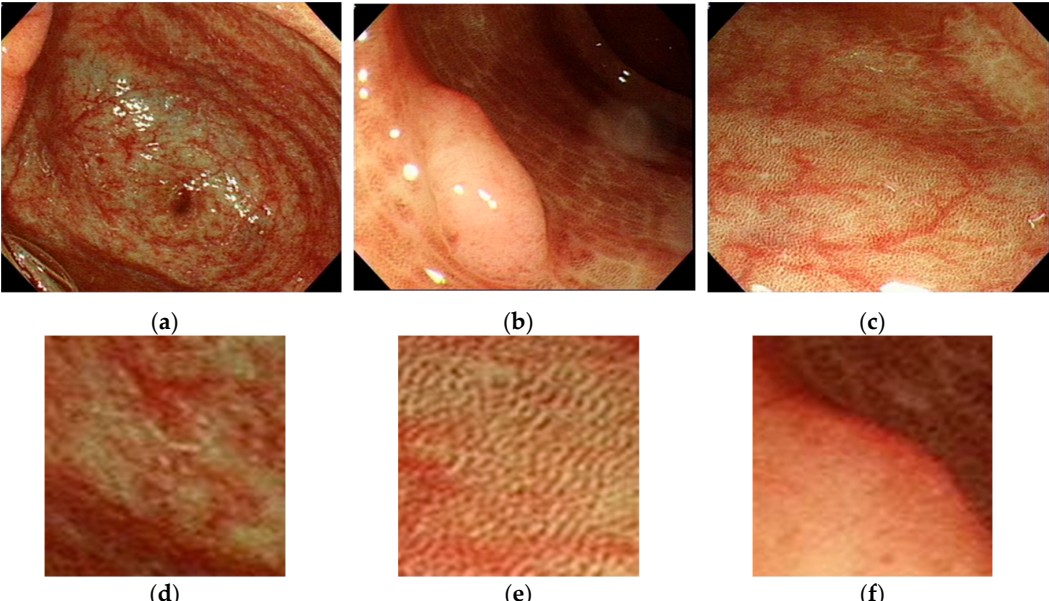

(a)    (b)    (c)

(d)    (e)    (f)

**Figure 2.** Images extracted from one patient with melanosis coli (**a**) cecum, (**b**) splenic flexure, and (**c**) colon polyp. The regions of interest (**d**) cecum—from (**a**); (**e**) splenic flexure—from (**b**); (**f**) colon polyp—from (**c**).

The images demonstrate that the MC has a particular dark brown pigmentation and presents a special black texture; places without pigmentation are white. The color textures of polyps or tumors are obviously different from the surroundings, exhibiting lighter and more turbulent textures, and most of them have no black texture.

We analyze the GLCM for quantitative feature extraction and the 14 features of the GLCM. The characteristics of the study are divided into three types: pattern, brightness, and texture features. Analysis focuses mainly on the texture features, and the results are calculated using the colonoscopy image RGB channels. The method for verifying the results uses the Pearson correlation coefficient to verify the correlation between features and pathological results. The tool calculates the Pearson correlation coefficients and $p$ values using MS Excel and IBM Statistical Package for the Social Sciences (SPSS). A Pearson correlation coefficient close to 0.4 and a $p$ of <0.05 represent significant correlation.

## 2.2. Image Texture Feature Analysis

Image texture features may be analyzed in many ways, with common methods including GLCMs, gray-level co-occurrence histograms, gray-level run length matrices, gray-level size zone matrices, neighboring gray tone difference matrices, gray-level dependence matrices, and etc. Although many new technologies have been developed so far, some traditional feature analysis methods still have their value. GLCMs are mainly used to count the probability of occurrence of pixel pairs in different directions and distances and our study focuses on similar textures on polyps and MC mucosa, so we selected GLCM method for this study.

The GLCM represents the grayscale value change of the relative positions for statistical pixels in space, reflecting the distribution of texture in space. The state was developed by Haralick et al. in 1973 [19]. Haralick believed that graphic texture is composed of multiple texture units and that the texture is caused by the repeated occurrence of the grayscale distribution in a spatial position. Therefore, a certain grayscale occurs between two pixels separated by a certain length in the image space. The spatial relationship of grayscale in an image can exhibit graphical characteristics [20]. The GLCM of an image can reflect information about the gray level of the image, such as that with respect to direction, adjacent interval, and variation amplitude. It constitutes the basis for analyzing the local patterns of images and their arrangement rules. In simple terms, data may be recorded in N×N square matrices according to grayscale relationships of pixels in images. For example, if the first lattice in a matrix is 1, only one pair of grayscales is 1 pixel horizontally adjacent. It is illustrated in Figure 3.

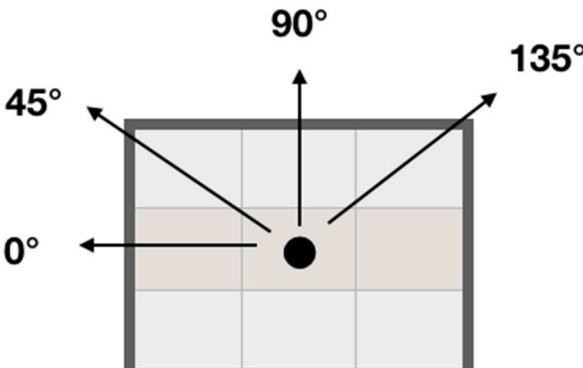

**Figure 3.** The gray-level co-occurrence matrices (GLCMs) features were extracted from four directions (0°, 45°, 90°, and 135°) with distance of 1 pixel from the centered pixel (●).

The GLCM has been widely used for texture analysis of various images types. Following various amendments, feature analysis methods were updated to improve results [21]. Ultrasound analysis for breast tumors [22] and rotator cuff tears [18], computed tomography for liver tumors [7] and hyperacute stroke [17], and image analysis for retinal vessels [23] have been successfully performed. The use

of computed tomography to simulate colonoscopy for polyp detection has also been studied [24]. In addition, one study analyzed pathological biopsies for colorectal cancer and normal colorectal mucosa [25], and this technique can provide clinicians with valuable assistance. Because this represents the earliest and the most mature image texture analysis method, this study analyzes the colorectal polyp image texture characteristics of patients with MC. The GLCM has a total of 14 features: autocorrelation, dissimilarity, energy, entropy, homogeneity, difference variance, difference entropy, information measure of correlation, inverse difference normalized, inverse difference moment, cluster prominence, cluster shade, contrast, and correlation. These features can be clinically applied to the analysis of various images. For the analysis of medical images, the features are employed mainly to study their correlations with various lesions. Of the 14 features analyzed in this study, only eight were related to the final results: entropy, energy, correlation, dissimilarity, homogeneity, autocorrelation, cluster_prominence, and cluster_shade.

Entropy is a measure of the quantity of information that an image possesses. Texture information also qualifies as image information. It is a measure of randomness. When all elements in the co-occurrence matrix possess maximum randomness and all values in the spatial co-occurrence matrix are almost equal, the elements in the co-occurrence matrix are dispersed, and the entropy (fluctuation) is large. This represents the degree of nonuniformity or complexity of the texture in the image.

Energy refers to the sum of the squares of the values of the GLCM elements, so it is also called energy. This reflects the uniformity of the gray scale distribution of the image and the texture thickness. If all values of the co-occurrence matrix are equal, the angular second moment (ASM) energy value is small; conversely, if some of the values are large and other values are small, the ASM energy is high. When the elements in the symbiotic matrix are concentrated, the ASM energy is high. Higher ASM energy indicates more uniform and regularly varying texture pattern.

Correlation is used to distinguish whether two objects have mutual correlations in shape and other features, and then, the correlation value is used to determine the characteristics of the object to locate the object. CorrelationM indicates the gray-level linear correlation between a pixel and its neighbors, similar to Correlation.

Dissimilarity is the degree of dissimilarity in gray-level value measurements for an image. It is sensitive to the arrangement of gray-level values in space or the hue of the image.

Homogeneity is used to reflect the homogeneity of image textures and to measure how much image texture changes locally. A large value indicates a lack of variation between different regions of the image texture, and the locality is largely uniform.

Autocorrelation is the degree of similarity of the metric spatial GLCM elements in the row or column direction. Therefore, the correlation value reflects the local grayscale correlation in the image. When the matrix element values are similar, the correlation value is large; conversely, if the matrix cell values differ greatly, the correlation value is small. If the image has a horizontal direction texture, the correlation value of the horizontal direction matrix is greater than the correlation value of the remaining matrix. The more vicious the image of the polyp, the higher the value of the relationship for horizontal or vertical textures is.

Cluster_prominence and cluster_shade indicate a lack of symmetry in the gray-level distribution. Therefore, the more malignant a polyp, the more complex is the polyp texture and the surrounding asymmetry.

In this study, the gastroenterologist manually used a $100 \times 100$-pixel box to mark the polyps then extracted and quantified the brightness features and texture features and analyzed the GLCM features of the colonoscopy images to obtain some of the MC. If the final pathological results confirm that texture features demonstrate Pearson correlation with pathological results, this model can be used as a reference for clinicians. As long as a gastroenterologist performs a colonoscopy, polyps that may be poorly pathologically differentiated can be treated immediately.

$$\text{Autocorrelation} = \sum_i \sum_j (p_x - \mu_x)(p_y - \mu_y)/\sigma_x \sigma_y \tag{1}$$

$$\text{Contrast} = \sum_n n^2 \left\{ \sum_i \sum_j p(i,j) \right\}, |i-j| = n \tag{2}$$

$$\text{Correlation} = \frac{\sum_i \sum_j (i-\mu_x)(j-\mu_y) p(i,j)}{\sigma_x \sigma_y} \tag{3}$$

$$\text{Cluster prominence} = \sum_i \sum_j (i+j-\mu_x-\mu_y)^4 p(i,j) \tag{4}$$

$$\text{Cluster shading} = \sum_i \sum_j (i+j-\mu_x-\mu_y)^4 p(i,j) \tag{5}$$

$$\text{Dissimilarity} = \sum_i \sum_j p(i,j)|i-j| \tag{6}$$

$$\text{Energy} = \sum_i \sum_j p(i,j)^2 \tag{7}$$

$$\text{Entropy} = -\sum_i \sum_j p(i,j) \log(p(i,j)) \tag{8}$$

$$\text{Homogeneity} = -\sum_i \sum_j \frac{1}{1+i-j} p(i,j) \tag{9}$$

$$\text{Difference variance} = \sum_i i^2 p_{x-y}(i) \tag{10}$$

$$\text{Difference entropy} = -\sum_i p_{x+y}(i) \log(p_{x+y}(i)) \tag{11}$$

$$\text{Information measure of correlation} = \frac{HXY-HXY1}{\max\{HX,HY\}} \; HXY = (8) \; HXY1 = -\sum_i \sum_j p(i,j) \log(p_x(i)p_y(j)) HX = \text{entropy of } p_x, HY = \text{entropy of } p_y \tag{12}$$

$$\text{Inverse difference normalized} = \sum_i \sum_j \frac{1}{1+|i-j|} p(i,j) \tag{13}$$

$$\text{Inverse difference moment} = \sum_i \sum_j \frac{1}{1+(i-j)^2} p(i,j) \tag{14}$$

where $\mu_x$, $\mu_y$, $\sigma_x$, and $\sigma_y$ are the mean and standard deviation (SD) of the marginal distributions of $p(i,j|d,\theta)$;

$$\mu_x = \sum_i i \sum_j p(i,j), \; \mu_y = \sum_j j \sum_i p(i,j) \tag{15}$$

$$\sigma_x^2 = \sum_i (i-u_x)^2 \sum_j p(i,j), \sigma_y^2 = \sum_j (j-u_y)^2 \sum_i p(i,j) \tag{16}$$

## 3. Results

Because some data came from the same patients at different ages, the age of the first discovery was taken as the patient's MC age. After removing duplicate patient medical records, we had 938 patients with MC diagnosed through colonoscopy in our hospital within six years; 694 (74%) were female and 244 (26%) male. The incidence of female patients after correction was 2.84 times higher than that of male patients. In total, 11 (1.2%), 341 (36.3%), 447 (47.7%), and 139 (14.8%) patients were aged <25, 26–50, 51–75, and >76 years, respectively. Therefore, age wise, the most MC-affected of patient group was of those aged >51 years; 62.5% of them were. Older patients were more likely to experience constipation. These patients are more likely to be affected by MC following anthranoid laxative use.

Among the 938 patients, 438 patients had CEA data. The CEA cutoff point was 4.7 ng/mL. Sixty-two people with CEAs exceeded the cutoff point, and compared the relationship between image texture GLCM and CEAs. After unclear and unsuitable patient data were excluded, 370 patients were finally analyzed.

First, the correlation between CEAs and MC mucosal texture was analyzed. For example, Pearson correlation between autocorrelation and CEAs was −0.005993648. Pearson correlation between contrast and CEAs was −0.045998786. Finally, no correlation occurred between CEAs and the texture characteristics of MC mucosa.

Among the 370 patients, 181 patients had polyps and pathological results, and the differentiative degree of polyps belonged to 130 of the adenomatous polyps. The ADR was 35.14% in 370 patients with MC. This result was similar to the 35.88% ADR for all 1092 individuals. Patients with MC exhibited an ADR similar to 34.7% cited by other studies in mainland China [1,26]. This also indicates that the presence or absence of CEA detection is not directly related to ADR in patients with MC, but patients with MC have a higher ADR than the general population does.

All the images were then analyzed using the GLCM for texture features. Because these 181 patients had polyps, each one had three type images for analysis. In total, 543 images were obtained. Table 1 presents the 181 patients' data and the SPSS calculations and statistics.

**Table 1.** Association between carcinoembryonic antigens (CEAs) and GLCMs texture features.

| Correlation (Number = 181) | | Correlation-M | Correlation | Dissimilarity | Energy |
|---|---|---|---|---|---|
| CEAs | Pearson | 0.094 | 0.094 | −0.073 | 0.001 |
| | Two-tailed significance | 0.210 | 0.210 | 0.331 | 0.991 |
| Correlation (number = 181) | | Entropy | Homogeneity-M | Homogeneity | |
| CEAs | Pearson | −0.068 | 0.088 | 0.087 | - |
| | Two-tailed significance | 0.361 | 0.237 | 0.244 | |

Continuing to count the polyp pathological results, we observed that the texture features and the classification of pathological results were related for certain characteristics. As discussed in other research, the pathological results of polyps in patients with MC are classified into three grades according to their benign and malignant degrees: The first grade includes benign hyperplastic polyp, Hamartoma polyp, pseudopolyps, and inflammatory polyps. The second involves adenomas, including tubular adenomas, tubulovillous adenomas, and villous adenomas. The third grade includes adenocarcinomas, along with both carcinoids and carcinomas in situ. If patients with MC have malignant colorectal adenocarcinomas, the textures of the endoscopy images are extremely disordered. The Pearson correlation coefficient between features and polyp grades was calculated.

First, the SPSS calculation demonstrated that the Pearson correlation coefficient was 0.390 ($p < 0.001$) for autocorrelation (Tables 2 and 3). It is illustrated in Figures 4 and 5.

**Table 2.** Association of autocorrelation with pathological grade.

| Correlation | | Autocorrelation |
|---|---|---|
| Polyp pathological grade (1–3) | Pearson correlation | 0.390 |
| | Two-tailed significance | $p < 0.001$ |
| | Number | 181 |

**Table 3.** Association of cluster_prominence with pathological grade.

| Correlation | | Cluster_Prominence |
|---|---|---|
| Polyp pathological grade (1–3) | Pearson correlation | 0.398 |
| | Two-tailed significance | $p < 0.001$ |
| | Number | 181 |

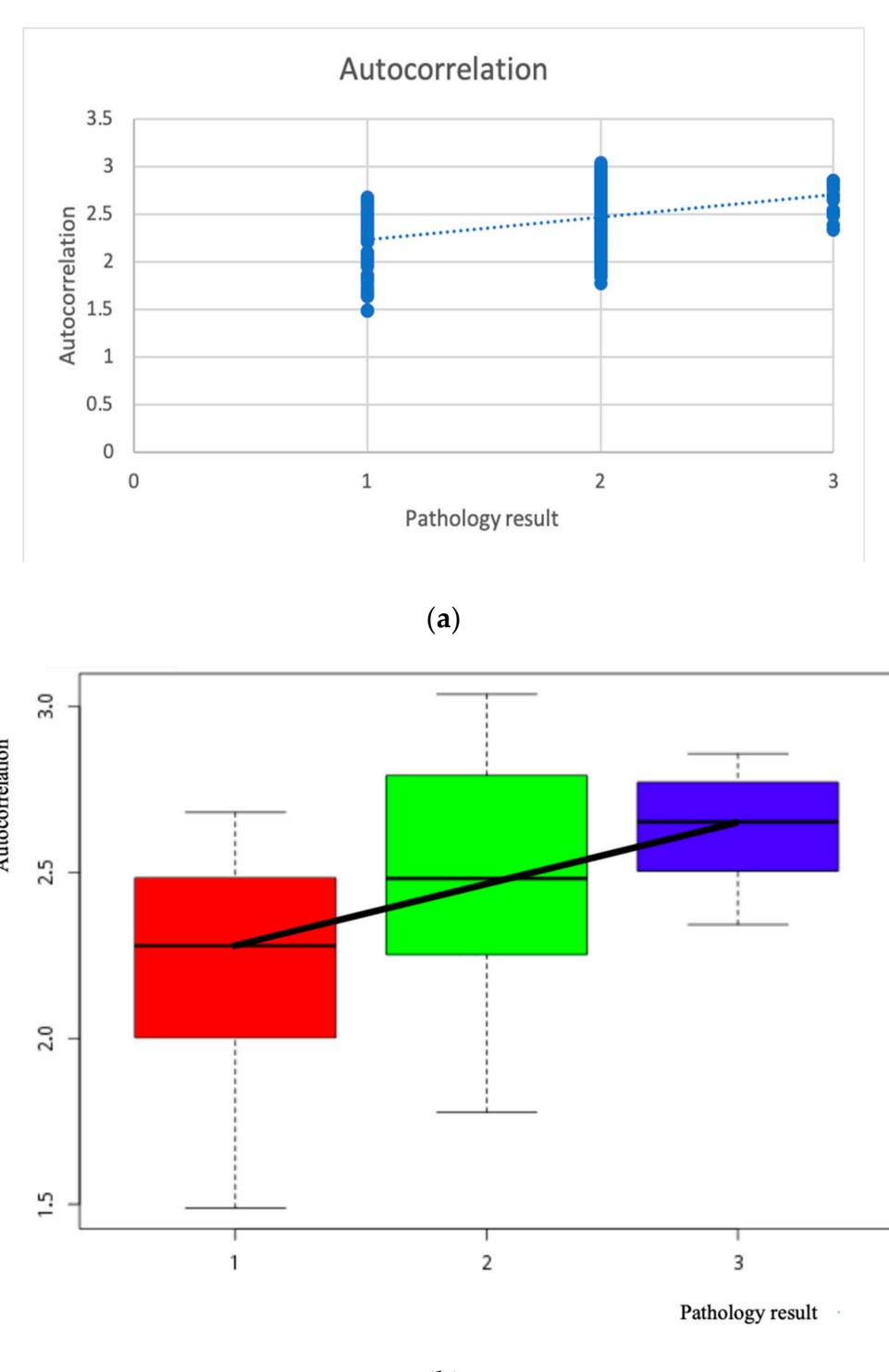

**Figure 4.** The correlations between autocorrelation and pathological results (**a**) scatter plot and (**b**) box-and-whisker plot. Pathology result 1 is hyperplastic polyps. Pathology result 2 is adenoma. Pathology result 3 is colon adenocarcinoma.

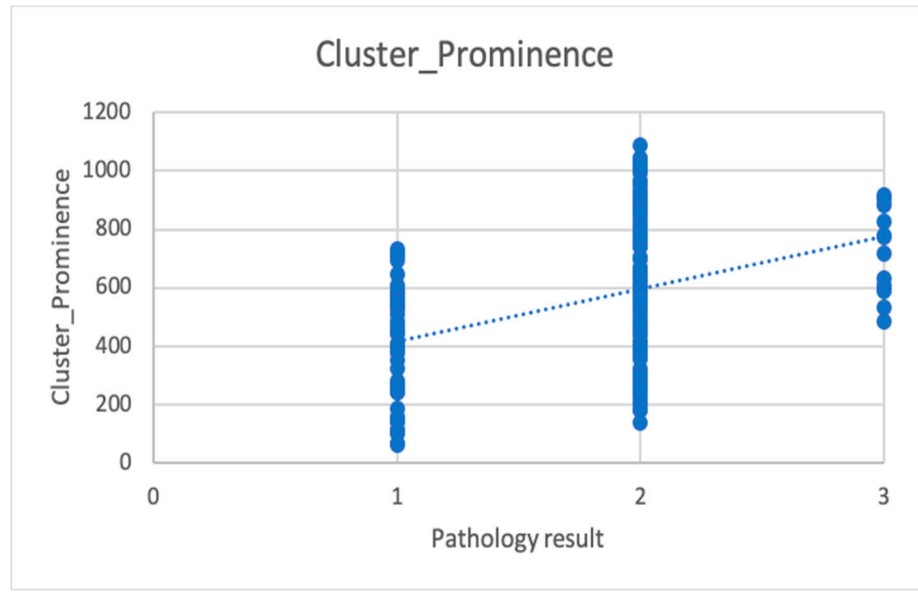

(**a**)

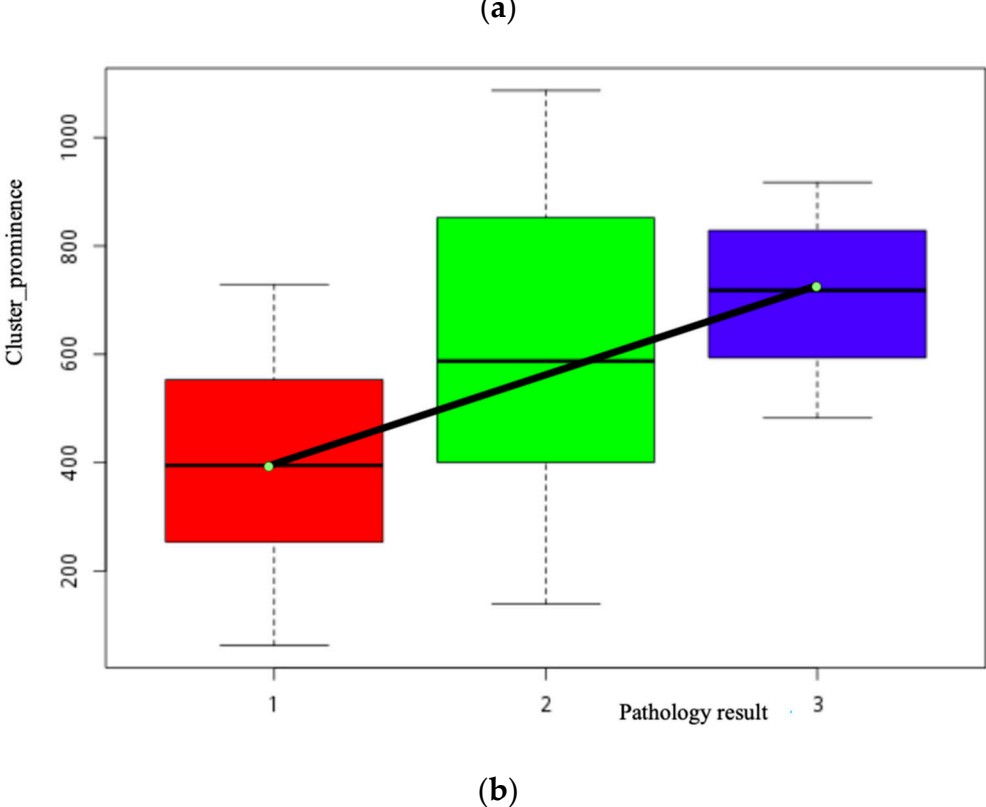

(**b**)

**Figure 5.** The correlations between cluster_prominence and pathological results (**a**) scatter plot and (**b**) box-and-whisker plot. Pathology result 1 is hyperplastic polyps. Pathology result 2 is adenoma. Pathology result 3 is colon adenocarcinoma.

For cluster_prominence and cluster_shade, the Pearson correlation coefficient was 0.398 ($p \leq 0.001$; Table 3, Figure 5) and 0.396 (slightly linear; $p$ value $\leq 0.001$; Table 4 and Figure 6) respectively.

**Table 4.** Association of cluster_shade with pathological grade.

| Correlation | | Cluster_Shade |
|---|---|---|
| | Pearson correlation | 0.396 |
| Polyp pathological grade (1–3) | Two-tailed significance | 0.000, $p < 0.05$ |
| | Number | 181 |

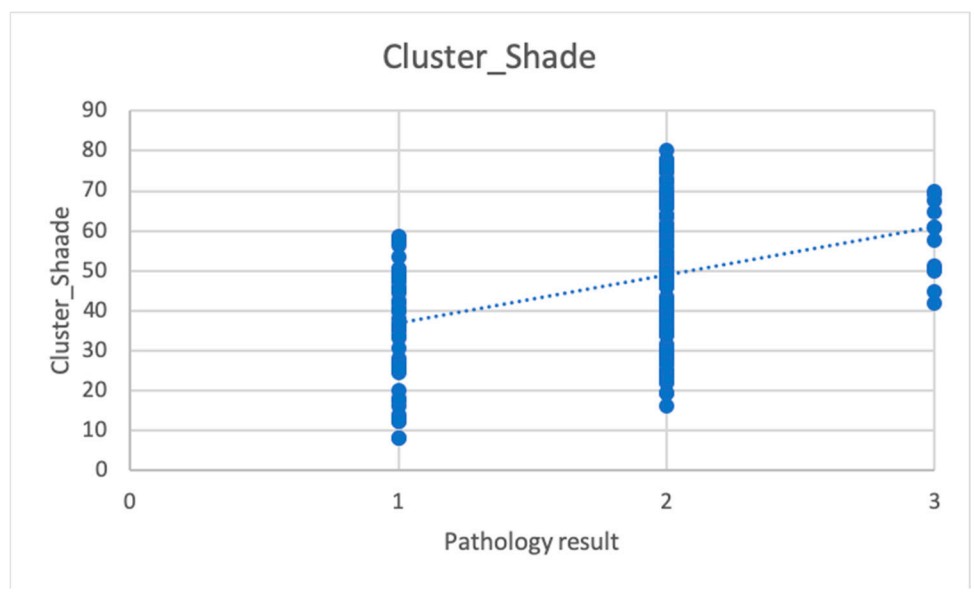

(**a**)

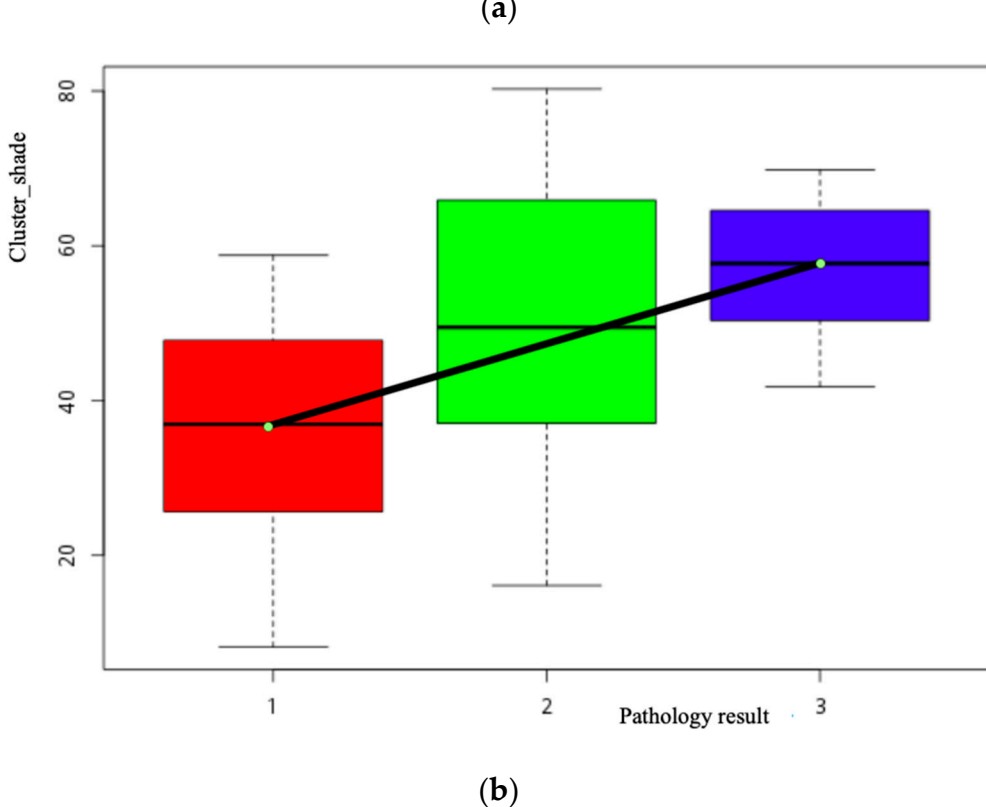

(**b**)

**Figure 6.** The correlations between cluster_shade and pathological results (**a**) scatter plot and (**b**) box-and-whisker plot. Pathology result 1 is hyperplastic polyps. Pathology result 2 is adenoma. Pathology result 3 is colon adenocarcinoma.

Finally, Table 5 presents SPSS results for total Pearson correlation between all GLCM features.

**Table 5.** All GLCM features and pathology correlation.

| Correlation Number = 181 | | Autocorrelation | Cluster_Prominence | Cluster_Shade |
|---|---|---|---|---|
| Polyp pathological grade (1–3) | Pearson correlation | 0.390 | 0.398 | 0.396 |
| | Two-tailed significance | $p < 0.001$ | $p < 0.001$ | $p < 0.001$ |

The features more related to the pathology result of polyps are autocorrelation, cluster_prominence, cluster_shade, and so on. These features can be considered for colonoscopic examinations, when MC and polyps are discovered, first for the gastroenterologist to predict possible pathological results.

Other features are furthermore weakly related to the pathological results, but the Pearson correlations are small. The *p* values are <0.05 in relation to, for example, contrast, correlationM, correlation, and difference_variance. This is illustrated in Table 6.

Contrast is slightly relevant to image texture. Contrast reflects image sharpness and texture depth. The deeper a texture groove, the greater the contrast and the clearer the visual effect. Conversely, if the contrast is small, the groove is shallow and the effect is blurred. The gray-level difference is the more pairs of pixels with a large contrast, the value is larger. The value of the element away from the diagonal in the GLCMs is larger, the contrast value. Therefore, if the type and time of laxative agent administration for patients with MC can be comprehensively recorded, longer duration of laxative agent use, patient has greater correlation of the patient's colon mucosa image and contrast. However, this aspect of the study requires more patient data, more time for data collection, and consent; further review by the School Ethics Committee is also required for more relevant research to be conducted.

**Table 6.** Summary of GLCMs and other pathological results.

| Correlation Number = 181 | | Contrast | CorrelationM | Correlation | Difference_Variance |
|---|---|---|---|---|---|
| Polyps pathological grade (1–3) | Pearson correlation | 0.266 | 0.235 | 0.235 | 0.266 |
| | Two-tailed significance | $p < 0.001$ | $p = 0.001$ | $p = 0.001$ | $p < 0.001$ |

## 4. Discussion

At the beginning of this study, we collected 298 patients' images and data in one-and-a-half years. The main purpose of this was to analyze the analysis mode of the larger quantity of data subsequently collected by analyzing smaller data quantities. After the preliminary image analysis, all image analysis data were quantified into an MS Excel table, and some texture features were demonstrated to exhibit Pearson correlation to CEAs. We observed that Pearson correlation between luminosity, chromaticity, and texture characteristics of the patient's colon mucosa did not differ significantly after correlation by patient's normal anal skin image data. The method of correcting the color of the patient's anal skin is not used. We used Image J software to cut images and to minimize interference from feces and reflections.

In addition, the study calculated that the ADR was 23.6% in the Division of Gastroenterology and Hepatology, Department of Internal Medicine, Taipei Medical University Hospital. The ADR of patients with MC was also calculated; it was higher in patients with MC higher in other patients, and the ratio of colorectal cancer was similar to the average rate in the general population.

Next, 1092 patient images were analyzed. Using the database to collect six years' worth of images and data, 543 patient images were ultimately enrolled for analysis. The correlation for GLCM analysis and CEAs has not improved, but we noted that patients with MC who had polyps also had five GLCM texture features correlated with pathological results. The three correlations were autocorrelation,

cluster_prominence, and cluster_shade; other features such as contrast, correlationM, correlation, and difference_variance also possess a certain degree of relevance. The five aforementioned texture features should be applicable to clinical colonoscopy image analysis.

No evidence currently suggests that MC is directly related to malignancies. But MC is related to adenoma [1] and adenoma is related to colorectal cancer [27]. To establish whether tumor markers (such as CEAs, CA199, CA125, AFP, or CA724) are related to texture features after calculation may require more data and image analysis studies. MC images require much clinical analysis, and the results should be provided for gastroenterologists prescribing relevant laxatives, mainly because MC is a reversible condition. For instance, if a patient who has MC with polyps exhibits particular textures or colon mucosal changes under colonoscopy, doctors can terminate the use of the relevant medication and prescribe different laxatives instead.

In addition, contrast mode reflects the clarity of the image and the degree of texture groove depth. The previous study did not mention that longer use of a relevant drug caused darker contrast. More data collection may confirm more correlations. Therefore, the longer drugs potentially able to affect colon mucosa coloring are used, the more obvious the texture. The effects of intestinal bacteria can be further analyzed in the future. In addition to GLCMs, other texture analysis methods such as gray-level run length matrix (16 features), gray-level size zone matrix (16 features), neighboring gray tone difference matrix (5 features), and gray-level dependence matrix (14 features) should be attempted. If these methods can be used to analyze polyp texture, new knowledge and understanding may be achieved. These quantitative image textures can also be combined in evaluating tumor malignancy or cancer types in the future using machine learning-based systems.

Some endoscopy studies have applied computer-assisted diagnostic analysis, such as capsule endoscopy, after capturing images of the intestines. The computer analysis of capsule endoscopes often entails use of MATLAB, mainly employing the red (R) channel primarily to capture red bleeding points or bleeding lesions [28]. However, colors other than red, such as black, green, and brown, are rarely included in the search. Therefore, the analysis of mucosal images and polyps under endoscopy is valuable [29]. The results of this study of MC suggest that in the future, endoscopy or colon-capsule endoscopy can provide clinical assistance in MC and polyp detection.

## 5. Conclusions

MC polyp texture was related to the pathological results of GLCM feature analysis, especially in the three features (autocorrelation, cluster_prominence, cluster_shade). In the future, photographs of the polyps taken during colonoscopy may potentially be used to detect colon MC and polyps and the five features for analysis employed to help gastroenterologists predict the possible pathological results for polyps in advance. This can save the patients' lives.

If this model can be developed, in the future, endoscopy doctors will be able to analyze uploaded colonoscopy images on a computer and establish the correlation between the texture features of different colorectal mucosa and clinical indicators, or even pathological results, then predict possible outcomes. In addition to colonoscopy, similar characterization methods should be applicable to other endoscopic imagery such as capsule endoscopic images. In short, this approach should be extensively applied in the future. This method effectively improves the efficiency of the examination and saves time, which is helpful for clinicians. This result should be used in the future to design an instant help software to help the gastroenterologist.

**Author Contributions:** Conceptualization, C.-M.L.; data curation, C.-M.L.; methodology, C.-M.L.; validation, C.-M.L.; writing—original draft, C.-M.L.; writing—review and editing, H.-J.Y.; investigation, H.-J.Y.; visualization, H.-J.Y.; supervision, C.-C.C. (Chun-Chang Chen); resource, C.-C.C. (Chun-Chao Chang); software, Y.-H.Y. All authors have read and agreed to the published version of the manuscript.

**Funding:** This research received no external funding.

**Acknowledgments:** The authors would like to thank the Division of Gastroenterology and Hepatology, Department of Internal Medicine, Taipei Medical University Hospital for financially supporting this research. This manuscript was edited by Wallace Academic Editor to assist with English grammar correction.

**Conflicts of Interest:** The authors declare no conflict of interest.

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
