# Peer review of "Quantitative Analysis of Melanosis Coli Colonic Mucosa Using Textural Patterns"

_applsci, doi:10.3390/app10010404_

Round 1

Reviewer 1 Report

The paper at first glance seems interesting, but already from the reading of the abstract, one realizes that the goal of the research is missing, unless the goal is, "The information of colonscopy can provide clinicians with suggestion for assessing patients with MC".

Continuing reading, I believe that the introduction is too long and also disperses the concepts that could be useful in understanding the goal.

 Why the authors exclude the data of patients without CEA? For sure CEA is the gold marker for colon cancer diagnosis and follow up, but in some case CEA has a false positive or false negative result [False-positive elevated CEA during colon cancer surveillance: a cholecystitis case report diagnosed by PET-CT scan Alireza Emami Ardekani, Hamidreza Amini, Zeinab Paymani, and Armaghan Fard-Esfahani] and [The diagnostic accuracy of carcinoembryonic antigen to detect colorectal cancer recurrence – A systematic review. Caspar G. Sørensen, William K. Karlsson, Hans-Christian Pommergaard, Jakob Burcharth, Jacob Rosenberg.  International Journal of Surgery Volume 25, January 2016, Pages 134-144], therefore the authors have lost a significant number of cases to be used in the retrospective analysis.

Different papers report that the pigmentation is evident only in microscopy [Nusko G, Schneider B, Müller G, et al. Retrospective study on laxative use and melanosis coli as risk factors for colorectal neoplasma. Pharmacology 1993; 47: 234-41.] or [. Kew ST. Melanosis coli. IeJSME 2012; 6 (Suppl 1): 553-60.] and the Endoscopy has sensitivity of 72% and specificity of 52% when compared to the reference method, which is histopathology [StÄ™pka M, Biernacka D, Górnicka B, et al. Melanoza jelita grubego– cechy charakterystyczne. Videochirurgia 2003; 8: 32-5] and [StÄ™pka M. Melanoza jelita grubego. Terapia 2003; 6: 50-2].

Which kind of suggestion can provide the single colonoscopy evaluation if not associated with histology evaluation too?

The presentation of results is very confused, and almost difficult to understand.

If the authors declare in the line 373 “No evidence currently suggests that MC is directly related to malignancies. The correlation between CEA, CA199, and texture features cannot be confirmed”, how they can support the hypothesis of the goal, that is  “The information of colonoscopy can provide clinicians with the suggestion for assessing patients with MC”?

In line 411 and 412, I believe the real goal of work is expressed. In the work of the clinician, the life of the patient must be present first and, the rapidity of evaluation must not serve to save time in the clinician work, but to save the patients’ life.

Author Response

Response to Reviewer 1 Comments

The paper at first glance seems interesting, but already from the reading of the abstract, one realizes that the goal of the research is missing, unless the goal is, "The information of colonscopy can provide clinicians with suggestion for assessing patients with MC".

Reply:

My opinion, the information on colonoscopy, can provide recommendations for clinicians to evaluate MC. I will be corrected in the abstract. This result should be used in the future to design an instant help software to help the physician.

Continuing reading, I believe that the introduction is too long and also disperses the concepts that could be useful in understanding the goal.

Reply:

I will shorten the introduction.

Why the authors exclude the data of patients without CEA? For sure CEA is the gold marker for colon cancer diagnosis and follow up, but in some case CEA has a false positive or false negative result [False-positive elevated CEA during colon cancer surveillance: a cholecystitis case report diagnosed by PET-CT scan Alireza Emami Ardekani, Hamidreza Amini, Zeinab Paymani, and Armaghan Fard-Esfahani] and [The diagnostic accuracy of carcinoembryonic antigen to detect colorectal cancer recurrence – A systematic review. Caspar G. Sørensen, William K. Karlsson, Hans-Christian Pommergaard, Jakob Burcharth, Jacob Rosenberg.  International Journal of Surgery Volume 25, January 2016, Pages 134-144], therefore the authors have lost a significant number of cases to be used in the retrospective analysis.

Reply:

In the initial analysis of the study, the analysis of polyp texture in patients without tumor index such as CEA or Ca199 was not related to pathological findings. The final analysis of patients with CEA is meaningful. It may be that these people are more likely to have cancer or polyps in the clinic and the clinician will measure CEA.

Different papers report that the pigmentation is evident only in microscopy [Nusko G, Schneider B, Müller G, et al. Retrospective study on laxative use and melanosis coli as risk factors for colorectal neoplasma. Pharmacology 1993; 47: 234-41.] or [. Kew ST. Melanosis coli. IeJSME 2012; 6 (Suppl 1): 553-60.] and the Endoscopy has sensitivity of 72% and specificity of 52% when compared to the reference method, which is histopathology [StÄ™pka M, Biernacka D, Górnicka B, et al. Melanoza jelita grubego– cechy charakterystyczne. Videochirurgia 2003; 8: 32-5] and [StÄ™pka M. Melanoza jelita grubego. Terapia 2003; 6: 50-2].

Which kind of suggestion can provide the single colonoscopy evaluation if not associated with histology evaluation too?

Reply:

  In current colonoscopy, Melanosis coli can be directly diagnosed and graded without waiting for tissue results.

The presentation of results is very confused, and almost difficult to understand.

If the authors declare in the line 373 “No evidence currently suggests that MC is directly related to malignancies. The correlation between CEA, CA199, and texture features cannot be confirmed”, how they can support the hypothesis of the goal, that is  “The information of colonoscopy can provide clinicians with the suggestion for assessing patients with MC”?

Reply:

  CEA and CA199 are not directly related to texture features, but polyps and texture features are associated in patients with CEA. MC is associated with adenomas, which are associated with malignancies. Therefore, if MC patients have more polyps and more adenomas, they are more likely to produce cancer in the future.

In line 411 and 412, I believe the real goal of work is expressed. In the work of the clinician, the life of the patient must be present first and, the rapidity of evaluation must not serve to save time in the clinician work, but to save the patients’ life.

Reply:

  Image assist can help clinicians determine multiple malignant lesions in the colon and treat them early, which can save the lives of patients.

Reviewer 2 Report

The proposed idea of “Quantitative Analysis of Melanosis Coli Colonic Mucosa Using Textural Patterns” seems interesting. Using Pearson correlation analysis based on correlations gray-level co-occurrence matrix and texture patterns, authors have analyzed the texture patterns in the colorectal images. This study presents an active approach and information of colonoscopy to clinicians with suggestions for assessing patients with MC. 

Though there are few issues that should be considered in the current research: 

1- The introduction section talks much more about the Melanosis coli (MC) and its background and patients data besides the proposed idea. My suggestion is to include a literature review or background section to present such information. 

 2- The Materials and Methods, section: Image texture feature analysis describes lots of definitions for the used methods, but there is no image, or mathematical analysis is provided to prove the proposed methods. The section Analysis of colonoscopic images is too short and offers no useful information on proposed methods and applications. 

3- The results section is having a good amount of tables and figures to present the finding. Though the tables are too big (sometimes going out of page dimensions) in size with data. Find some suitable ways to present table data. Graphs are poorly labeled and present no description of the given results. For example, Table 2, Graph B. supposed to have a connection between columns 1 2 and 3. The same problem with Table 3 and 4 graphs. 

4- And off course there more need of patients' data to prove the authenticity of the proposed method. 

5- From Reference section Reference 1-5 (most in middle and end) presented the doi number is that necessary if it is why not for the rest of Reference. The reference style is not standard. 

6- Suggestion: these days, machine learning-based systems are doing well for detection and diagnoses of medical images. I hope you will consider such options in future studies.  

Author Response

Response to Reviewer 2 Comments

The proposed idea of “Quantitative Analysis of Melanosis Coli Colonic Mucosa Using Textural Patterns” seems interesting. Using Pearson correlation analysis based on correlations gray-level co-occurrence matrix and texture patterns, authors have analyzed the texture patterns in the colorectal images. This study presents an active approach and information of colonoscopy to clinicians with suggestions for assessing patients with MC.

Though there are few issues that should be considered in the current research:

1- The introduction section talks much more about the Melanosis coli (MC) and its background and patients data besides the proposed idea. My suggestion is to include a literature review or background section to present such information.

Reply:

I will shorten the introduction.

 2- The Materials and Methods, section: Image texture feature analysis describes lots of definitions for the used methods, but there is no image, or mathematical analysis is provided to prove the proposed methods. The section Analysis of colonoscopic images is too short and offers no useful information on proposed methods and applications.

Reply:

  Image texture feature analysis is to analyze the image of the colonoscopy image and analyze it with a computer program. So there is no image in this part.

3- The results section is having a good amount of tables and figures to present the finding. Though the tables are too big (sometimes going out of page dimensions) in size with data. Find some suitable ways to present table data. Graphs are poorly labeled and present no description of the given results. For example, Table 2, Graph B. supposed to have a connection between columns 1 2 and 3. The same problem with Table 3 and 4 graphs.

Reply:

The article will be corrected according to your instructions and suggestion.

4- And off course there more need of patients' data to prove the authenticity of the proposed method.

Reply:

I will collect more data later.

5- From Reference section Reference 1-5 (most in middle and end) presented the doi number is that necessary if it is why not for the rest of Reference. The reference style is not standard.

Reply:

Sorry, the errors in the references will be corrected.

6- Suggestion: these days, machine learning-based systems are doing well for detection and diagnoses of medical images. I hope you will consider such options in future studies.

Reply:

Thanks for your advice.

Round 2

Reviewer 1 Report

The manuscript could be accepted for the publication

Author Response

Response to Reviewer 1 Comments

1-The paper at first glance seems interesting, but already from the reading of the abstract, one realizes that the goal of the research is missing, unless the goal is, "The information of colonscopy can provide clinicians with suggestion for assessing patients with MC".

Response 1:

Thank you for your comments. I aggred with your suggestion. The goal of introduction will change to "The information of colonscopy can provide clinicians with suggestion for assessing patients with MC".The information on colonoscopy can provide recommendations for clinicians to evaluate MC. I will be corrected in the abstract. This result should be used in the future to design an instant help software to help the physician.

2-Continuing reading, I believe that the introduction is too long and also disperses the concepts that could be useful in understanding the goal.

Response 2:

According to your suggestions, we have shorten the introduction to focus on the main concepts for readers to understand the goal.

3-Why the authors exclude the data of patients without CEA? For sure CEA is the gold marker for colon cancer diagnosis and follow up, but in some case CEA has a false positive or false negative result [False-positive elevated CEA during colon cancer surveillance: a cholecystitis case report diagnosed by PET-CT scan Alireza Emami Ardekani, Hamidreza Amini, Zeinab Paymani, and Armaghan Fard-Esfahani] and [The diagnostic accuracy of carcinoembryonic antigen to detect colorectal cancer recurrence – A systematic review. Caspar G. Sørensen, William K. Karlsson, Hans-Christian Pommergaard, Jakob Burcharth, Jacob Rosenberg.  International Journal of Surgery Volume 25, January 2016, Pages 134-144], therefore the authors have lost a significant number of cases to be used in the retrospective analysis.

Response 3:

In the initial analysis of the study, the analysis of polyp texture in patients without tumor index such as CEA or Ca199 was not related to pathological findings. The final analysis of patients with CEA is meaningful. These people are more likely to have cancer or polyps in the clinic and the clinician will measure CEA.

4-Different papers report that the pigmentation is evident only in microscopy [Nusko G, Schneider B, Müller G, et al. Retrospective study on laxative use and melanosis coli as risk factors for colorectal neoplasma. Pharmacology 1993; 47: 234-41.] or [. Kew ST. Melanosis coli. IeJSME 2012; 6 (Suppl 1): 553-60.] and the Endoscopy has sensitivity of 72% and specificity of 52% when compared to the reference method, which is histopathology [StÄ™pka M, Biernacka D, Górnicka B, et al. Melanoza jelita grubego– cechy charakterystyczne. Videochirurgia 2003; 8: 32-5] and [StÄ™pka M. Melanoza jelita grubego. Terapia 2003; 6: 50-2].Which kind of suggestion can provide the single colonoscopy evaluation if not associated with histology evaluation too?

Response 4:

In current colonoscopy, melanosis coli can be directly diagnosed and graded without waiting for tissue results. In the past, endoscopes could not achieve better or the equivalent of microscopic evaluation may be the limitation of human vision or it is difficult to generalize the presentation of different tissue colors. Therefore, this study proposes the texture analysis of image processing to find the correlation and reduce the dependence on the microscope. An endoscopic observation can immediately have a quantitative assessment recommendation

The presentation of results is very confused, and almost difficult to understand.

5-If the authors declare in the line 373 “No evidence currently suggests that MC is directly related to malignancies. The correlation between CEA, CA199, and texture features cannot be confirmed”, how they can support the hypothesis of the goal, that is  “The information of colonoscopy can provide clinicians with the suggestion for assessing patients with MC”?

Response 5:

We made the results become more clearer by expressing the following statement. Polyps and texture features are associated with pathology result in patients. MC is associated with adenomas, which are associated with malignancies. Therefore, if MC patients have more polyps and more adenomas, they are more likely to produce cancer in the future. We will corret this article.

6-In line 411 and 412, I believe the real goal of work is expressed. In the work of the clinician, the life of the patient must be present first and, the rapidity of evaluation must not serve to save time in the clinician work, but to save the patients’ life.

Response 6:

Thank you for your comments.

Image assist can help clinicians determine multiple malignant lesions in the colon and treat them early, which can save the lives of patients in the future.

Reviewer 2 Report

- In the earlier review, I have mentioned some major formate flaws. The reviewed version still contains those issues. 

 Table 1, 5, and 6 go outside text margins.

- The problem I mentioned earlier regards graphs is still there. 

- No signification improvement has shown in terms of experiments, methods, or techniques. 

Author Response

Response to Reviewer 2 Comments

As attached file below

Round 3

Reviewer 2 Report

The authors have made significant progress after multiple review. Now it seems that the current paper is suitable for publication after a few formate changes.